# Standard Nutritional Assessment Tools Are Unable to Predict Loss of Muscle Mass in Patients Due to Undergo Pancreatico-Duodenectomy: Highlighting the Need for Detailed Nutritional Assessment

**DOI:** 10.3390/nu16091269

**Published:** 2024-04-25

**Authors:** Mary E. Phillips, M. Denise Robertson, Kate Bennett-Eastley, Lily Rowe, Adam E. Frampton, Kathryn H. Hart

**Affiliations:** 1Department of Nutrition and Dietetics, Royal Surrey NHS Foundation Trust, Guildford GU2 7XX, UK; 2Faculty of Health and Medical Sciences, University of Surrey, Guildford GU2 7XH, UK; 3HPB Surgical Unit, Royal Surrey NHS Foundation Trust, Guildford GU2 7XX, UK; 4Section of Oncology, Department of Clinical and Experimental Medicine, FHMS, University of Surrey, Guildford GU2 7XH, UK

**Keywords:** hand-grip strength, pancreatico-duodenectomy, malnutrition, post-operative outcomes, prehabilitation, GLIM criteria

## Abstract

Background and Methods: Pancreatico-duodenectomy (PD) carries significant morbidity and mortality, with very few modifiable risk factors. Radiological evidence of sarcopenia is associated with poor outcomes. This retrospective study aimed to analyse the relationship between easy-to-use bedside nutritional assessment techniques and radiological markers of muscle loss to identify those patients most likely to benefit from prehabilitation. Results: Data were available in 184 consecutive patients undergoing PD. Malnutrition was present in 33–71%, and 48% had a high visceral fat-to-skeletal muscle ratio, suggestive of sarcopenic obesity (SO). Surgical risk was higher in patients with obesity (OR 1.07, 95%CI 1.01–1.14, *p* = 0.031), and length of stay was 5 days longer in those with SO (*p* = 0.006). There was no correlation between skeletal muscle and malnutrition using percentage weight loss or the malnutrition universal screening tool (MUST), but a weak correlation between the highest hand grip strength (HGS; 0.468, *p* < 0.001) and the Global Leadership in Malnutrition (GLIM) criteria (−0.379, *p* < 0.001). Conclusions: Nutritional assessment tools give widely variable results. Further research is needed to identify patients at significant nutritional risk prior to PD. In the meantime, those with malnutrition (according to the GLIM criteria), obesity or low HGS should be referred to prehabilitation.

## 1. Introduction

Pancreaticoduodenectomy (PD) carries a 40% morbidity and 0.5–3.8% mortality but is the only option for cure in those with peri-ampullary tumours [1,2]. Centralisation of surgical services has resulted in a dramatic reduction in post-operative mortality [1,3]. Thus, research has focused on the identification of potentially modifiable risk factors to ensure patients not only survive the surgery but maintain a performance status that results in tolerance of adjuvant chemotherapy, as the combination of these treatments is associated with the best long-term survival [4].

Prehabilitation is recognised as an appropriate intervention for those at high surgical risk, and successful prehabilitation interventions are associated with improvements in muscle function [5]. However, the decision to delay potentially curative cancer surgery for a period of prehabilitation is a difficult one, and there is an urgent need to identify markers that predict poor surgical outcomes and thus allow a better selection of patients for prehabilitation. 

Over the last 10 years, there have been dramatic changes in the methods available for nutritional assessment, as well as in our understanding of the impact malnutrition has on surgical outcomes [6,7]. Loss of muscle mass, measured radiologically, is associated with higher rates of surgical complications and associated mortality [8,9,10] and has been the only potentially modifiable risk factor identified so far [10]. 

Sarcopenia, the loss of muscle mass and muscle function, can occur in patients who are obese and weight stable, meaning standard nutritional screening tools currently used at pre-assessment cannot reliably detect sarcopenia, as these focus primarily on body weight. 

Historically, surgical risk has been based on body mass index and weight loss, but in PD, the presence of radiologically measured low muscle mass in patients who fall into the clinically obese category (classified as sarcopenic obesity) is associated with a higher risk of pancreatic fistula and with “failure to rescue”, meaning that not only are patients with sarcopenic obesity more likely to experience a life-threatening complication but, if they do, they are less likely to survive it [10]. 

At present, we do not screen patients for sarcopenia, and thus, these patients will not be referred to prehabilitation. The current gold standard for muscle mass assessment is the analysis of dual energy X-rays (DXA) [11] or the quantity and quality of muscle at the L3 (third part of the lumbar spine) region on a computerised tomography (CT) scan [12], the latter being used more frequently as clinically indicated CT images can be utilised [13]. These radiological measures should be used in conjunction with a functional assessment to confirm the presence of sarcopenia. However, this is time-consuming and requires specialist software, so there is a significant drive to validate a cheap, quick, portable and easily accessible alternative. 

To date, there is no data exploring the link between cheap and readily available assessments of muscle function (hand-grip strength—HGS) and nutritional assessment tools with radiological measurements of muscle mass using CT. 

Whilst changes in HGS are accepted as a marker of change in muscle function and predict survival in non-surgical cohorts [14], there is no data on the association between single-point measurements and muscle mass. The aim of our study was to evaluate the link between HGS and radiological evidence of sarcopenia in patients due to undergo PD. HGS assessment is a quick and easy test, and if validated, could be used to predict the need for prehabilitation.

## 2. Materials and Methods

A retrospective review of CT images was undertaken using Slice-o-matic (version 5, Tomovision, Magog, QC, Canada), and these were compared to surgical complications, length of stay, nutritional markers and hand-grip strength from a prospectively collated database of consecutive patients undergoing pancreatic head resection maintained between 2014 and 2018. On internal review, this study was classified as a clinical audit due to the retrospective nature of data collection; therefore, formal ethical opinion was not required. However, clinical audit approval was received (reference DCSS-CA-22-23-047). All patients who underwent PD in 2014–2018 were included, with no exclusion criteria. 

### 2.1. Technique and Quality Assessment

CT images were uploaded into Slice-o-matic, and the sagittal images were used to select the axial image in the centre of L3. The Alberta protocol [15] was used for analysis and completed by hand. 

Two operators carried out repeated analysis on a set of ten images until their results were reproducible to <3% error. Screenshots were taken at each step of the process, and these were assessed by a third operator for accurate selection of the L3 image (Figure 1) and completion of the assessment of skeletal muscle (SM), visceral adipose tissue (VAT), subcutaneous adipose tissue (SAT), and intramuscular adipose tissue (IAT) (Figure 2). 

Tissue types were defined by the Hounsfield unit (HU) as skeletal muscle (SM) −29 to 150 HU, visceral adipose tissue (VAT) −150 to −50 HU, and intramuscular adipose tissue (IMAT) and subcutaneous adipose tissue (SAT) as −190 to −30 HU [16]. The mean HU for the muscle analysis was collated to assess muscle quality. 

The skeletal muscle index was calculated by dividing the skeletal muscle mass (cm^2^) by height^2^. Sarcopenia was determined when the skeletal muscle area (SMA) or skeletal muscle index (SMI) was below the 5th percentile of normal. Poor muscle quality was determined when the muscle radiation attenuation (MRA) was below the 5th percentile (Table 1) [17].

Sarcopenic obesity was defined as those with a low skeletal muscle index and a BMI > 30 kg/m^2^ or as a ratio of VFA/SMI with a cut-off of 2.5 m^2^ [18].

### 2.2. Hand-Grip Strength

HGS measurements were taken in the pre-assessment clinic using a Takei Hand Grip Strength dynamometer (Takei Scientific Instruments Co., Lft, Japan). Subjects stood for the assessment with their arms straight by their side. Each measurement was taken three times in both their dominant and non-dominant hands. Patients were encouraged to relax their hands for 1 minute between each measurement. 

HGS was analysed in three ways. Firstly, by selecting the mean of the three measurements; secondly, by taking the highest measurement; and finally, by quantifying the difference (as a percentage change) between the first and third measurements. We hypothesised this latter assessment may be a marker for muscle fatigue. 

### 2.3. Nutritional Markers

Body weight, percentage weight loss in three months prior to surgery, body mass index, and the malnutrition universal screening tool (MUST) data were collected at pre-assessment. Patients with a MUST score of 2 or more were defined as at “nutritional risk” [19].

The European Society for Clinical Nutrition and Metabolism (ESPEN) malnutrition definitions [20] were applied retrospectively with those with a BMI < 20, grip strength < 85% normal for their age and sex, or >5% weight loss defined as meeting the phenotypic criteria. Patients with reduced food intake and clinical signs of malabsorption or inflammation met the etiologic criteria [20]. Those with at least one in each category were defined as malnourished. 

The severity of malnutrition was defined in line with the ESPEN guidelines as follows: -“moderate malnutrition” if there was 5–10% weight loss in 6 months; 10–20% overall weight loss; a BMI < 20 kg/m^2^ for those under 70 (or <22 kg/m^2^ for those ≥70 years old) or mild to moderate muscle weakness (low HGS). -“Severe malnutrition” if there was >10% weight loss in the last 6 months; >20% overall weight loss; a BMI < 18.5 kg/m^2^ in those under 70 years old (or <20 kg/m^2^ in those ≥70 years old) or severe muscle function deficit (low HGS) [20].

### 2.4. Surgical Outcomes

Length of hospital stay, 30-day mortality, and surgical complications were collated. The Clavien–Dindo scoring system was used to grade the severity of post-operative complications. A Clavien–Dindo score ≥3 was considered a significant complication [21]. The International Study Group for Pancreatic Surgery (IPGS) classifications were used to denote the severity of delayed gastric emptying, post pancreatectomy haemorrhage, and pancreatic fistula [22,23,24], with clinically relevant pancreatic fistula (CRPF) defined as a grade B or C fistula [23]. 

### 2.5. Statistical Analysis

Demographic data were assessed for normality using Shapiro–Wilk tests. Mann–Whitney tests or independent samples *t*-test were used to compare outcomes between variables. Binary outcomes were analysed using chi-squared and logistic regression. 

Pearson’s correlations were used to look at the strength of the relationship between nutritional factors, using the definition of +/−<0.3 for negligible correlation, +/−0.3–0.5 for a low correlation, +/−0.5–0.7 for a moderate correlation, +/−0.7 to 0.9 for high correlation, and +/−0.9–1.00 for a very high correlation [25]. Logistic regression was performed in those with a moderate or strong correlation to determine any predictive values. 

Analyses were carried out in SPSS (Version 28, IBM, Armonk, NY, USA) and Stata (Version 16, StataCorp, 2019, College Station, TX, USA).

## 3. Results

### 3.1. Demographics

Data were analysed on one hundred and eighty-four consecutive patients undergoing pancreatico-duodenectomy (PD). One hundred and two (55.4%) were male, and the mean age was 65.1 years (SD 10.5). One hundred and fifty-two patients underwent a pylorus-preserving PD (PPPD), twenty-three a Full Whipple (PD with distal gastrectomy), and four a total pancreatectomy. Five patients had a PD with additional surgical procedures, which were liver resection (*n* = 2) or, vascular resection (*n* = 2) or hemi colectomy (*n* = 1) (Table 2).

Histology was malignant in 70% of cases, premalignant in 16% of cases, and benign in 14% of cases (Table 2).

### 3.2. Complications

Three patients were admitted pre-operatively for nutritional support (parenteral nutrition, *n* = 2; enteral nutrition, *n* = 1). 

Five patients required a re-laparotomy (Table 3), and 31 patients (17%) had a Clavien–Dindo (CD) score of 3 or above. Pancreatic fistula was present in 57 cases; delayed gastric emptying was present in 33 cases. Bile leak was present in 8 cases, 16 patients had a chyle leak, 12 patients had a post-operative bleed, 27 had a post-operative infection, and 8 patients went home on nasojejunal tube feeding (Table 3).

Other complications with a CD score ≥ 3 were ICU admissions for CPAP [for exacerbation of COPD (*n* = 1), chest infection (*n* = 1), pleural effusion (*n* = 1)]; inotrope requirement post-op (*n* = 2), cardiac failure (*n* = 1), amiodarone infusion (*n* = 1), and leukocyte adhesions deficiency (LAD) syndrome (*n* = 1).

### 3.3. Nutritional Assessment

CT images were available in 124 cases (67%), with the remaining imaging unavailable as it had been accessed from other trusts remotely, and these electronic links were no longer available. Both GS and CT images were available in 97 patients (52%). 

MUST scores were available in 173 cases (94%). The mean MUST score was 1.2 (SD 1.48), and 32.9% were deemed at risk of malnutrition (MUST score ≥ 2) (Table 4).

Weight loss in the 3 months preceding surgery was available for 173 patients. Mean weight loss was 6.86% (SD 7.1). Twenty-nine per cent (*n* = 54) did not report any weight loss, 23% (*n* = 40) reported less than 5% weight loss, 55% (*n* = 96) more than 5% weight loss, 25% (*n* = 44) reported 5–10% weight loss, and 31% (*n* = 52) had more than 10% weight loss. 

The mean body mass index (BMI) was 25.1 kg/m^2^ (SD 4.14), 10% (*n* = 18) had a BMI less than 20 kg/m^2^, and 10% (*n* = 17) had a BMI over 30 kg/m^2^. Using the presence of sarcopenia and BMI > 30, only one patient met the criteria for sarcopenic obesity, whereas when using the ratio of VFA/SMI, 57 patients (31%) had a radio of ≥2.5, suggestive of sarcopenic obesity. Using this category, there was a significant increase in LOS in those with sarcopenic obesity (*p* = 0.006) (Table 4).

Using the ESPEN GLIM criteria, severe malnutrition was present in 43% of cases and moderate malnutrition in 28%. GS was available in 138 cases (75%) but not clinically indicated in five patients. GS was deemed not clinically indicated due to emotional distress (n = 2), severe arthritis (n = 1), tremor (*n* = 1), and residual weakness from polio (*n* = 1). The median HGS (dominant hand) was 26.7 kg (SD 8.69) and 24.5 kg (SD 9.0). The BMI was lower in those with sarcopenia (22.4 ± 3.5 compared to 25.8 ± 4.0, *p* < 0.001) (Table 4).

Overall, the incidence of malnutrition varied from 10% to 71%, depending on the tool used (Figure 3). Only one patient did not score on any radiological, functional or nutritional marker. 

### 3.4. Outcome Data

The median length of stay was 10 days (IQR 7–17) overall, and there was no significant difference between MUST categories (*p* = 0.339) or between those considered at nutritional risk (MUST ≥ 2) or not (MUST < 2), (*p* = 0.691). 

Whilst patients who were sarcopenic, had obesity, had an HGS < 85% normal, or had low muscle quality all had a higher complication rate than the overall population (20% or above), this only reached significance for those with obesity (*p* = 0.039) (Table 4). 

Skeletal muscle index (SMI) was associated with clinically relevant pancreatic fistula (CRPF), with those with lower muscle mass more likely to have CRPF (*p* = 0.003).

A high BMI was associated with all complications (OR 1.16, 95%CI 1.07–1.27, *p* = 0.001), pancreatic fistula (OR 1.13, 95%CI 1.04–1.22, *p* = 0.004) and infection (OR 1.16, 95%CI 1.05–1.28, *p* = 0.003). Still, this data is limited by the lack of control of confounding variables.

Infectious complications were associated with low SMI (*p* = 0.031), the high percentage change in HGS in the dominant hand (*p* = 0.008), and approached significance with low muscle quality (*p* = 0.054) and VFA/SMI ratio (*p* = 0.058). Infectious complications were associated with a high MUST score (*p* < 0.001) and the presence of malnutrition according to the GLIM criteria (*p* < 0.001).

Pearson’s correlations were used to look for associations between SMI or muscle quality and nutritional assessment markers. Weak correlations were found between SMI and BMI (0.457, *p* < 0.001) and HGS (mean dominant measurement: 0.468, *p* < 0.001; highest dominant measurement: 0.472, *p* < 0.001; highest non-dominant measurement: 0.431, *p* < 0.001). There was a weak negative correlation between the presence of malnutrition determined by the GLIM criteria and SMI (−0.379, *p* < 0.001). Muscle quality correlated weakly with the highest HGS in both the dominant (0.445, *p* < 0.001) and non-dominant hand (*p* = 0.321, *p* < 0.001) (Table 5).

## 4. Discussion

The aim of this study was to evaluate the use of HGS as a marker for increased surgical risk and the need for prehabilitation. Our data demonstrates the need for complete assessment and confirms that whilst only weakly correlated, HGS was a stronger marker for predicting low muscle stores than nutritional assessment tools, such as MUST, which is widely used in clinical practice. 

Assessment of nutritional status is complex, and nutritional screening and assessment tools are diverse. Nutritional screening tools are designed to identify patients at risk of malnutrition, whilst nutritional assessment tools are designed to determine nutritional status. These terms are often used interchangeably, resulting in confusing data. 

A study retrospectively applying 12 different nutritional risk tools to 279 patients prior to pancreatic surgery found no correlation between any of the nutritional risk tools and surgical outcomes. The incidence of malnutrition detected by these tools varied wildly, from 1.1 to 79.6% [26].

ESPEN redefined nutritional assessment [20] to allow for the inclusion of muscle stores or functions, highlighting the need to include this within the nutritional assessment. However, there remains limited data regarding the impact of a change in muscle function prior to surgery and its implications on the outcome. 

Within our cohort, the incidence of malnutrition varied from 33% using MUST, 55% using 5% weight loss, and 71% with the GLIM ESPEN criteria, the most common tools available in the United Kingdom. Further work should include patient-generated data such as the Patient Generated Subjective Glo Assessment tool (PGSGA), which includes an assessment of the ability to undertake activities of daily living as well as nutritional impact symptoms [27,28], thus providing a more holistic tool and has been used in conjunction with CT images in other cohorts [29].

The definition of sarcopenia varies considerably between studies, but it is accepted to involve a combination of muscle mass and strength [30]. Although early work associated the term sarcopenia as an effect of ageing, the impact of poor physical activity, malnutrition, and many other factors is widely recognised [30]. The GLIM criteria incorporate HGS as a marker of muscle function, setting a standard of >85% of normal, allowing for age, gender, and BMI [14,20], although there is no sensitivity or specificity data available [31]. In contrast to previous work on DXA images, CT imaging is now considered the gold standard for quantification of muscle mass, predominantly due to availability [31].

Single-point radiological assessments of muscle stores have been used in a number of studies and suggest not only radiological sarcopenia [8,32] but specifically sarcopenic obesity is associated with poor outcomes [18,33,34]. We were able to replicate the association between radiological sarcopenia and CRPF (*p* = 0.003) but were not able to replicate the sarcopenic obesity association due to the low mortality within our cohort (*n* = 2).

Other studies have identified sarcopenia as a risk factor for increased complication rates (OR 4.3; 95%CI 2.2–8.5, *p* < 0.01) and longer LOS (*p* < 0.05), and on multivariate analysis have identified SMA, a low BMI, and male gender as predictors of increased LOS [35]. 

In a retrospective review of 333 patients undergoing PD, patient demographics revealed a heavier population with a BMI reported as 27.9 ± 5.5 kg/m^2^ in the non-sarcopenic cohort and 25.9 ± 4.4 kg/m^2^ in those with sarcopenia [35], compared to a BMI of 25.9 ± 4.0 kg/m^2^ in our non-sarcopenic group and 22.4 ± 3.5 kg/m^2^ in our sarcopenic group. In contrast to this study, we identified a higher complication rate in those with a high BMI (*p* = 0.039), but this data is limited due to a lack of control of potentially confounding variables. Malnutrition identified with MUST, GLIM, or hand-grip strength was associated with infectious complications, which occurred in 15% of patients. 

Very large-scale studies reported a similar BMI to our cohort median of 25.6 kg/m^2^ (IQR 23.0–28.3) but identified that 16.6% of the cohort were obese, and 21.3% of those with obesity also had sarcopenia [9]. In our study, the incidence of obesity was low at only 10%, and within that, only one patient with obesity also had low muscle mass within our study (*n* = 1). Using the proposal by Ryu et al. of a radio between visceral fat area to skeletal muscle index to determine sarcopenic obesity [18], increased the rate of radiological SO to nearly 50% of the cohort, highlighting the need for standardised assessment tools. 

### 4.1. Hand-Grip Strength

Many studies have explored the impact of HGS on outcome. Studies using a single time point and mean HGS have demonstrated that HGS was independently associated with hospital mortality in mechanically ventilated patients [36], and readmissions in patients with intestinal failure on home parenteral nutrition [37]. Further studies identified low HGS as an independent predictor of treatment reduction in oesophageal cancer [38] and increased length of stay in 109 patients undergoing liver transplant [39], 130 cancer patients [40], and 108 patients having major abdominal surgery [41]. 

Traditionally, HGS has been analysed as a mean of three readings, compared to a reference range specific to age and sex [42]. However, repeated tests over time are considered more clinically useful, and there is a wide variation in baseline measurements, as demonstrated by our median HGS of 26.7 and 24.5 kg with standard deviation of 8.6 and 9.0 for dominant and non-dominant hands retrospectively. Our hypothesis was that the change in HGS from the first to the last result would represent muscle fatigue and be more clinically useful than a single result was not confirmed in this study, as there was a negligible correlation between these markers and SMI or muscle quality. 

HGS has been assessed in 137 patients prior to pancreaticoduodenectomy and was low in 42.3% of patients [43]. Further work in a small study (*n* = 26) noted HGS reduced significantly from baseline up to 3 months post-op from a median of 27 to 19.3 (*p* < 0.001) and demonstrated a significant reduction in all nutritional markers, highlighting the need for nutritional support in the post-operative setting [44], but neither of these studies correlate nutritional markers with the outcome.

The link between nutritional status and surgical outcomes is difficult to define due to the high number of operative variables. Delayed gastric emptying is primarily associated with operative variables [45], and chyle leaks are heavily associated with the rate of lymphatic dissection [46]. Pancreatic fistula is associated with soft glands, small pancreatic ducts, and fat content of the gland [47]. Given the number of variables, further work attempting to quantify the impact of nutritional factors on multivariate analysis should be undertaken in a multicentre study. 

There is limited data exploring the correlation between functional assessments. A literature search identified only one study of 112 healthy volunteers that explored correlations between muscle mass assessments using DXA and bioelectric impedance (BIA), muscle function using HGS, “timed up and go”, and jump power with biochemical markers such as creatine (methylD3 dilution). They found HGS and jump power correlated with both DXA and BIA, but biochemical markers and “timed up and go” did not [48].

We attempted to correlate skeletal muscle index with a measure of HGS to ascertain if this tool could be used to identify patients at risk of surgical complications. However, we only found weak correlations between the highest HGS in both dominant and non-dominant hands. Additionally, we did not identify any strong correlations between measures of muscle mass and quality and any of the HGS measures. Given the incorporation of HGS within the GLIM criteria, it is likely that the presence of HGS within the tool is the reason why the GLIM criteria were weakly negatively correlated with SMI. Therefore, HGS alone may be sufficient to predict risk.

### 4.2. Limitations and Areas for Future Research

The retrospective nature of this study meant that it was not possible to apply further functional assessment or nutritional risk tools that require biochemical assessment. However, the combination of GLIM criteria and the MUST tool reflects current clinical practice in the United Kingdom.

Whilst surgical risk factors are an important outcome, one could argue that the most clinically relevant outcomes are long-term survival, specifically supporting patients undergoing the combination of surgery and adjuvant chemotherapy. Therefore, future work should explore the risk factors for and impact of prehabilitation on the ability to commence and tolerate adjuvant chemotherapy, and this work could be carried out in other surgical cohorts. 

HGS can be influenced by factors other than muscle strength. Underlying medical conditions will impact the interpretation of HGS, and this is demonstrated within this study, where severe arthritis, the residual effects of polio as a child, and neuropathy have a direct impact on HGS. Studies in elderly care have also demonstrated the impact of anaemia on functional tests, including HGS [49]. Further studies should include data on potential confounders such as anaemia. 

The definitions of sarcopenia are different depending on the population. For this study, we used data determined from studies of healthy Caucasians aged from 20 to 82 years old [17]. However, whilst the study was carried out in a predominantly Caucasian population, ethnicity was not collected during the study, and there may be some participants for whom different reference ranges would be more appropriate. 

Furthermore, when considering further studies, the definition of prehabilitation varies. Systematic reviews combine data from studies analysing the impact of exercise alone with those incorporating nutritional assessment, endocrine and exocrine optimisation, and psychological support [50].

Future studies would benefit from the addition of a third data point between diagnosis and surgery, which would allow for the study of the impact of individual and multi-modal interventions to help design the optimal prehabilitation for different surgical cohorts. 

### 4.3. Implications for Practice

This study highlights the wide variety of assessment tools and conflicting data within the literature. Of note, only one patient did not score on any assessment tools for malnutrition, poor muscle mass, or poor muscle function. Our hypothesis that muscle fatigue with deteriorating hand-grip strength might be a clinically useful marker of reduced muscle mass was not confirmed. Given the variation in assessment tools and outcomes and the positive impact of prehabilitation in both length of stay and morbidity [50], further work should assess which patients benefit from prehabilitation. This assessment could involve using rehabilitation targets and the ability to complete adjuvant chemotherapy as long-term markers of success. In the meantime, all patients with evidence of malnutrition, obesity, or loss of muscle mass measured as low hand-grip strength should be referred to prehabilitation services. 

Our cohort had a lower incidence of obesity compared to other published studies, and thus, our data may not be applicable to other populations with a higher incidence of obesity. 

In the absence of readily available assessment of muscle mass, HGS and the GLIM criteria should be used in preference to MUST or percentage weight loss to identify patients with low muscle mass. However, the development of alternative tools is needed.

## 5. Conclusions

Twenty-four per cent of our population demonstrated radiological evidence of sarcopenia. However, there was a high incidence of malnutrition in our cohort, with a wide variation depending on the tool used, in line with other published studies (33–71%). 

We did not see a relationship between skeletal muscle index or muscle quality with the MUST assessment tools or simple weight loss equations, but there was a weak correlation with the ESPEN GLIM criteria and measures of HGS. Further work should explore the impact of HGS and other functional assessment tools on surgical outcomes directly and, subsequently, the impact of intervention on outcomes. In the meantime, patients at risk of malnutrition, with low HGS or obesity, should be referred to prehabilitation services prior to PD. 

## Figures and Tables

**Figure 1 nutrients-16-01269-f001:**
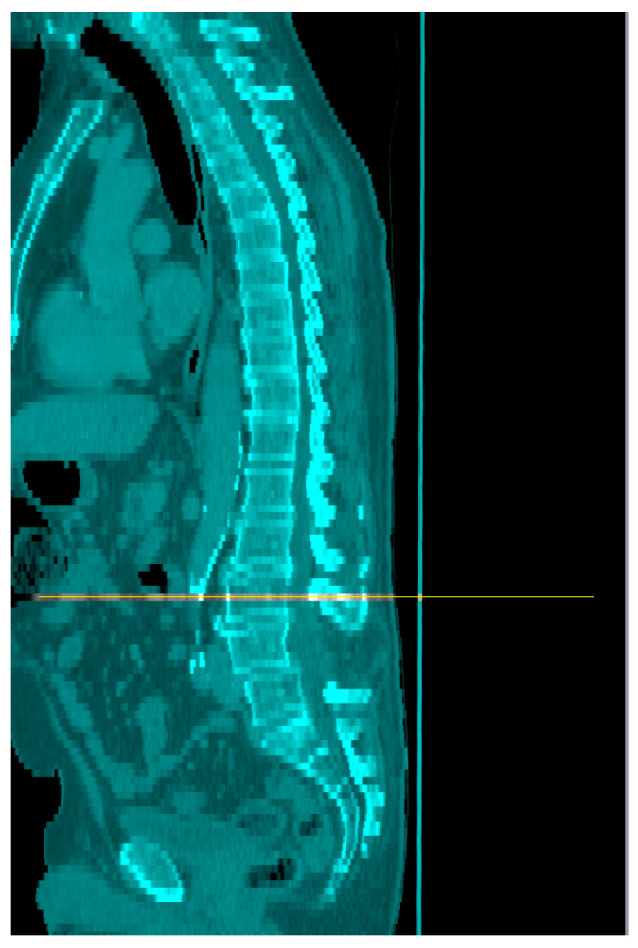
Identification of the mid-point of L3 on sagittal CT images (orange line).

**Figure 2 nutrients-16-01269-f002:**
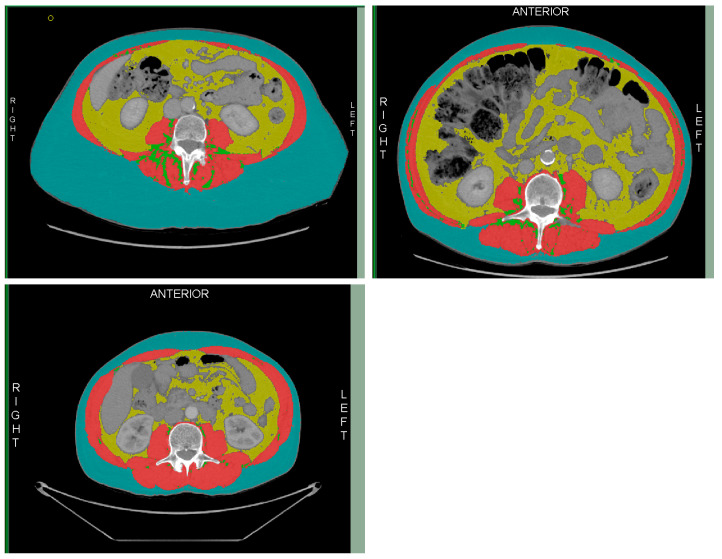
Division of skeletal muscle (red); intramuscular adipose tissue (green); visceral adipose tissue (yellow); and subcutaneous adipose tissue (blue).

**Figure 3 nutrients-16-01269-f003:**
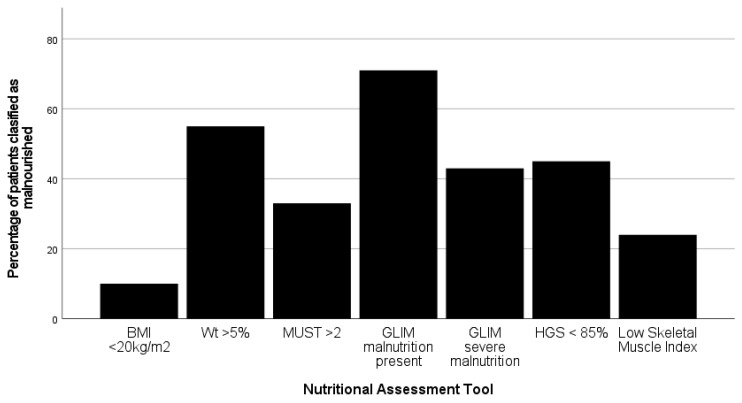
Variation in the incidence of malnutrition when the seven different nutritional assessment tools are applied to 173 patients undergoing pancreatico-duodenectomy. (BMI—body mass index; Wt—weight; MUST—malnutrition universal screening tool; GLIM—global leadership initiative on malnutrition; HGS—hand-grip strength).

**Table 1 nutrients-16-01269-t001:** Cut-offs at the 5th percentile were used to define sarcopenia and poor muscle quality [17].

	Males	Females
Skeletal muscle area (cm^2^)	134.0	89.2
Skeletal muscle index (cm^2^/m^2^)	41.6	32
Muscle radiation attenuation (HU)	29.3	22

**Table 2 nutrients-16-01269-t002:** Demographics and pre-operative nutritional assessment in 184 consecutive patients undergoing pancreatico-duodenectomy.

Demographics		Pre-Operative Nutritional Assessment	
Patient demographics (*n* = 184)		MUST score (*n* = 173)	
Male gender	102 (55%)	Mean MUST score	1.2 (SD 1.48)
Mean age (SD)	65.1 (10.5)	MUST ≥ 2	57 (33%)
Operation type		MUST < 2	116 (67%)
Pancreatico-duodenectomy	152 (83%)	BMI (*n* = 179)	
Full Whipple	23 (13%)	BMI < 20 kg/m^2^	21 (10%)
Total pancreatectomy	4 (2%)	BMI > 20 kg/m^2^	158 (90%)
Additional procedures		BMI > 30 kg/m^2^	21 (10%)
Liver resection	2	BMI < 30 kg/m^2^	158 (90%)
Vascular resection	2	HGS (*n* = 132)	
Hemi-colectomy	1	HGS < 85% normal	59 (45%)
Histology		HGS > 85% normal	73 (55%)
Pancreatic ductal carcinoma	67 (36%)	GLIM criteria (*n* = 153)	
Cholangiocarcinoma	25 (14%)	Severe malnutrition	66 (43%)
Cancer of the ampulla	20 (11%)	Not severe malnutrition	87 (57%)
Chronic pancreatitis	18 (10%)	Malnutrition	107 (71%)
Intraductal papillary mucinous neoplasm	15 (8%)	No evidence malnutrition	46 (29%)
Tubular villous adenoma	13 (7%)	CT analysis (*n* = 122)	
Neuroendocrine tumours	8 (4%)	Low skeletal muscle index	29 (24%)
Cancer of the duodenum	8 (4%)	Normal skeletal muscle index	91 (76%)
Bile duct stones/cholangitis	4 (2%)	High fat to muscle mass ratio (VFA/SMI)	57 (48%)
Familial adenomatous polyposis	2	Poor muscle quality	15 (12%)
Brunner’s gland hamartoma	1	Normal muscle quality	106 (88%)
Autoimmune pancreatitis	1	Weight loss (*n* = 173)	
Gangliocytic paraganglioma	1	>5% weight loss	96 (55%)

(MUST—malnutrition universal screening tool; SD—standard deviation; BMI—body mass index; HGS—hand-grip strength; GLIM—global leadership initiative on malnutrition; CT—computerised tomography).

**Table 3 nutrients-16-01269-t003:** Surgical outcomes in 184 consecutive patients undergoing pancreatico-duodenectomy.

Surgical Outcome			
Complications (*n* = 184)		Other complications	
30-day/90-day mortality	2 (1.1%)/2 (1.1%)	Home with NJ feeding	8 (4%)
All Clavien–Dindo scores ≥ 3	31 (17%)	Infectious complication	27 (15%)
Re-lapatorotomy	5 (2.7%)	Chyle leak	16 (9%)
Pancreatic fistula		Post pancreatectomy haemorrhage	12 (7%)
All	57 (31%)	Bile leak	8 (4%)
Grade A	21 (11%)	Parenteral nutrition related complications	
Grade B	30 (16%)	All	4 (2%)
Grade C	6 (3%)	Line sepsis	2
Clinically relevant pancreatic fistula	36 (19%)	Electrolyte imbalance	1
Delayed gastric emptying		Hypertriglyceridaemia	1
All	33 (18%)	Length of stay	
Grade A	5 (3%)	Median (IQR)	10 days (7–17)
Grade B	18 (10%)		
Grade C	10 (5%)		

(NJ—nasojejunal; IQR—interquartile range).

**Table 4 nutrients-16-01269-t004:** Difference in length of stay and severe complications between different categories of nutritional risk in 184 consecutive patients undergoing pancreatico-duodenectomy.

Risk Score	Risk Score Present	Mean Length of Stay (SD)	Mann–Whitney	Clavien–Dindo Complications ≥ 3	Chi-Squared
Whole population		14.17 (12.0)		31/185 (17%)	
MUST score (*n* = 173)					
MUST ≥ 2	57 (33%)	15.09 (11.7)	*p* = 0.160	8/57 (14%)	*p* = 0.281
MUST < 2	116 (67%)	13.9 (12.6)		22/116 (19%)	
Body mass index (BMI) (*n* = 179)					
BMI < 20 kg/m^2^	21 (10%)	12.86 (11.2)	*p* = 0.915	2/21 (10%)	*p* = 0.271
BMI > 20 kg/m^2^	158 (90%)	14.42 (12.3)		28/158 (18%)	
BMI > 30 kg/m^2^	21 (10%)	20.55 (19.2)	*p* = 0.131	7/21 (33%)	*p* = 0.039
BMI < 30 kg/m^2^	158 (90%)	13.48 (10.8)		23/158 (15%)	
Hand-grip strength (HGS) (*n* = 132)					
HGS < 85% normal	59 (45%)	13.41 (12.7)	*p* = 0.905	13/60 (22%)	*p* = 0.114
HGS > 85% normal	73 (55%)	12.59 (9.4)		9/73 (12%)	
GLIM criteria (*n* = 153)					
Severe malnutrition	66 (43%)	13.85 (11.0)	*p* = 0.870	9/66 (14%)	*p* = 0.287
Not severe malnutrition	87 (57%)	14.12 (12.9)		16/87 (18%)	
Malnutrition	107 (71%)	14.44 (12.6)	*p* = 0.477	17/108 (16%)	*p* = 0.463
No evidence malnutrition	46 (29%)	12.96 (10.7)		8/45 (18%)	
CT analysis (*n* = 121 for skeletal muscle index, *n* = 122 for muscle quality)					
Low skeletal muscle index	29 (24%)	12.31 (10.4)	*p* = 0.385	6/28 (21%)	*p* = 0.333
Normal skeletal muscle index	91 (76%)	14.2 (13.0)		14/92 (15%)	
Sarcopenia and BMI > 30 kg/m^2^	1	Not suitable for analysis			
VFA/SMI ≥ 2.5 (Sarcopenic obesity)	57 (48%)	16.01 (14.7)	*p* = 0.006	13/60 (22%)	*p* = 0.391
VFA/SMI < 2.5	61 (52%)	11.27 (9.2)		9/58 (16%)	
Poor muscle quality	15 (12%)	13.73 (6.6)	*p* = 0.179	3/15 (20%)	*p* = 0.498
Normal muscle quality	106 (88%)	13.8 (13.0)		18/107 (17%)	

(MUST—malnutrition universal screening tool; SD—standard deviation; BMI—body mass index; HGS—hand-grip strength; GLIM—global leadership initiative on malnutrition; CT—computerised tomography; VFA—visceral fat area; SMI—skeletal muscle index).

**Table 5 nutrients-16-01269-t005:** Correlations between nutritional assessment tools and skeletal muscle index or muscle quality in 97 patients at pre-operative assessment before pancreatico-duodenectomy.

Nutritional Assessment Tool	Skeletal Muscle Index	Muscle Quality
	Correlation Co-Efficient	
BMI	0.457 (*p* < 0.001)	−0.273 (*p* = 0.003)
Percentage weight loss	−0.261 (*p* = 0.004)	−0.167 (*p* = 0.69)
MUST score	−0.262 (*p* = 0.005)	−0.098 (*p* = 0.291)
GLIM criteria for malnutrition present	−0.379 (*p* < 0.001)	−0.192 (*p* = 0.036)
Hand-grip strength—dominant hand		
Mean	0.468 (*p* < 0.001)	0.284 (*p* = 0.005)
Per cent of normal	0.207 (*p* = 0.049)	0.085 (*p* = 0.42)
Highest	0.24 (*p* = 0.022)	0.445 (*p* < 0.001)
Percentage Difference	−0.297 (*p* = 0.659)	0.047 (*p* = 0.659)
Absolute Difference	−0.093 (*p* = 0.382)	0.084 (*p* = 0.423)
Hand-grip strength—non-dominant		
Mean	0.097 (*p* = 0.350)	0.108 (*p* = 0.294)
Per cent of normal	−0.033 (*p* = 0.755)	−0.112 (*p* = 0.288)
Highest	0.431 (*p* < 0.001)	0.321 (*p* = 0.002)
Percentage Difference	−0.107 (*p* = 0.328)	0.00 (*p* = 0.998)
Absolute Difference	−0.130 (*p* = 0.224)	−0.056 (*p* = 0.6)

(BMI—body mass index; MUST—malnutrition universal screening tool; GLIM—global leadership initiative on malnutrition).

## Data Availability

The original contributions presented in the study are included in the article, further inquiries can be directed to the corresponding author.

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
