# Peer review of "Standard Nutritional Assessment Tools Are Unable to Predict Loss of Muscle Mass in Patients Due to Undergo Pancreatico-Duodenectomy: Highlighting the Need for Detailed Nutritional Assessment"

_nutrients, 2024, doi:10.3390/nu16091269_

Round 1

Reviewer 1 Report

Comments and Suggestions for Authors

It is a well-written paper with an area of improvement in Methods. 

1. Please check the correctness of the mortality rate and its reference on the first page under the Introduction.

2. Materials and Methods: The research team reviewed the CT images retrospectively. They collected data on surgical complications, lengths of stay, nutritional markers, and hand grip strength prospectively. Please clarify the time frame of the retrospective CT images and the timeline for the prospective data collection. In addition, please provide the sample information, including inclusion/exclusion criteria and recruitment methods.

3. Please provide information about the ethical considerations.

Author Response

Thank you for your review, and constructive comments to help us improve our paper. I have added additional references, which are clearer, to the mortality as the data were difficult to pull from the website initially referenced - thank you for highlighting this. 

I have also added a line regarding the clinical audit approval we had and the time frame data were collected over.

Data on all patients who underwent pancreaticoduodenectomy between 2014-2018 were included, there were no exclusion criteria. I have expanded on our methodology to explain this. 

Thank you. 

Reviewer 2 Report

Comments and Suggestions for Authors

I found the paper very interesting and well described The information is clear and provided appropriately I think the concept should be stressed that this methodology, although not easy to apply, can be used in different categories of patients

I would try to compare this method with other sarcopenia conditions and give a small message of nutritional strategy with impact on effectiveness and improvement of sarcopenia 

Author Response

Thank you for your comments to help us strengthen our paper. I have added the potential to use this methodology in other cohorts and to assess effectiveness of interventions for sarcopenia. 

Thank you